# GROUPED CORRELATION AGGREGATION WITH PROPAGATION FOR STEREO MATCHING

## ABSTRACT

Iterative optimization-based methods have dominated the field of stereo matching with extraordinary precision and speed. However, these methods still suffer from low iteration efficiency and insufficient correlation volume with low utilization rates. As the countermeasure, we propose *grouped correlation aggregation with propagation*, *aka.*, GCAP-STEREO, a novel stereo matching method inspired by traditional methods. We design an efficient updater to improve the performance of single iteration optimization. To alleviate the problems of correlation volume, a novel grouped window shifting mechanism and a contour-aware aggregation modified from semi-global matching (SGM) have been introduced. Our method outperforms all methods in zero-shot generalization and ranks 1$^{st}$ on ETH3D among published works. Additionally, we conducted targeted inference optimization on the video stream and demonstrated the improvement in frame rate without sacrificing accuracy through experiments on the simulator. Finally, a real-world binocular system is deployed to qualitatively demonstrate the practicality of our method.

## 1 INTRODUCTION

Stereo matching is a vital task in computer vision that has numerous practical fields such as 3D reconstruction, autonomous driving, and AR/VR (Jamiy & Marsh, 2019; Fan et al., 2018). It aims to obtain the pixel-level matching relationships between two images captured by the calibrated binocular system, namely the disparity.

Many popular traditional methods have demonstrated significant effectiveness in both theory and practice. Semi-Glocal Matching (SGM) (Hirschmüller, 2005) uses mutual information to evaluate matching cost, and approximates a global two-dimensional smoothing constraint by aggregating one-dimensional constraints. PatchMatch Stereo (Bleyer et al., 2011) randomly initializes disparity, then propagates and optimizes disparity between pixels, gradually obtaining a high-quality disparity map. However, due to the lack of parallelism and insufficient perception of image information, these methods cannot meet the real-time and accuracy requirements in practical scenarios.

With the development of deep neural networks, learning-based methods have demonstrated absolute advantages in the field. CNN-based methods such as PSMNet (Chang & Chen, 2018) use an amount of convolutions to complete information extraction and the matching cost calculation. These methods has improved computational efficiency and accuracy, but they still cannot meet the requirements for high pixel-level tasks due to the high memory and computing power demands. Moreover, the iterative optimization-based method adopts a storage instead of computation approach to store the matching cost volumes between all pixels of images and uses lightweight convolutional recurrent neural network (convRNN) units for iterative updates. It can balance accuracy and time by dynamically setting the number of iterations, and the memory overhead is also relatively small.

However, there are still some points to consider in optimization-based methods. Firstly, a single iteration update is not efficient enough and the methods require a sufficient number of iterations to achieve the required accuracy. Secondly, the matching cost volume only considers the matching cost between single pixels, which leads to frequent occurrences of noise points and matching errors. Finally, the methods store the matching relationships of all pixels, but a large part of the relationships are not accessed at all and occupy memory.

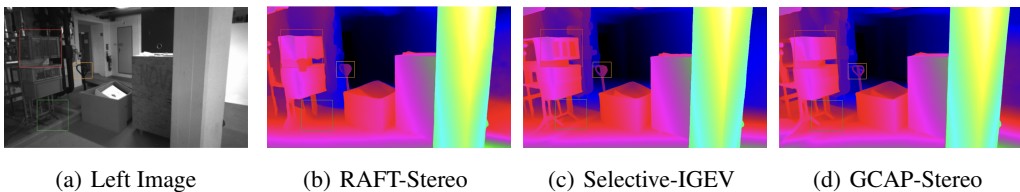

(a) Left Image     (b) RAFT-Stereo     (c) Selective-IGEV     (d) GCAP-Stereo

Figure 1: Examples of our predictions ETH3D benchmark with RAFT-Stereo (Lipson et al., 2021) and Selective-IGEV (Wang et al., 2024). Our method is particularly outstanding in areas with holes and weak textures.

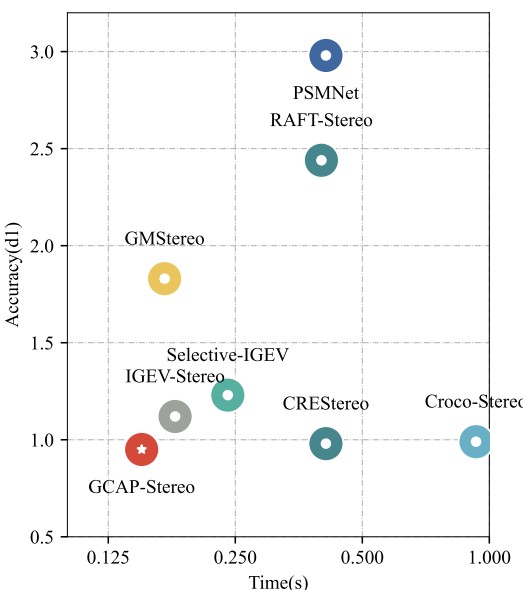

Figure 2: Comparisons with state-of-the-art stereo methods on ETH3D leaderboards

In this paper, we propose GCAP-Stereo, namely grouped correlation aggregation with propagation to deal with the above considerations. We design a new iterative updater based on PatchMatch Stereo (Bleyer et al., 2011) to improve the single quality of optimization. Moreover, a modified SGM cost aggregation (Hirschmüller, 2005) has been utilized in the cost volume. Considering the situation where there are a large number of matching costs that will not be accessed, we introduce the grouped window-shifting mechanism to retain all valid points and discard the vast majority of invalid points. Finally, we perform targeted inference optimization on video streams to achieve higher frame rates in practical scenarios without affecting the accuracy.

So far, GCAP-Stereo ranks 1st on ETH3D two-view stereo (Schöps et al., 2017) benchmarks and achieves competitive performance on KITTI 2012/2015 (Geiger et al., 2012) and Middlebury (Scharstein et al., 2014) among published methods. As shown in fig. 2, our method is ahead of all other methods in terms of speed and accuracy. Moreover, our method demonstrates excellent performance advantages in video stream testing and zero-shot generalization, which has superior cross-domain generalization and real-time performance. Our main contributions can be summarized as follows:

- We design a novel updater based on PatchMatch Stereo for iterative stereo matching methods which improves the single optimization performance.
- We propose a modified SGM-based cost aggregation to improve the robustness of cost volume with little time consumption.
- We introduce the grouped window-shifting mechanism to greatly reduce the cost volume and decrease the probability of using incorrect matching points.
- Our method outperforms existing methods on public benchmarks such as ETH3D and demonstrates advantages in zero-shot generalization and video stream inference.

## 2 RELATED WORK

### 2.1 TRADITIONAL METHODS

Stereo matching is a fundamental issue and there are many crucial research achievements. Traditional methods generally consist of several steps, including matching cost calculation, cost aggre-

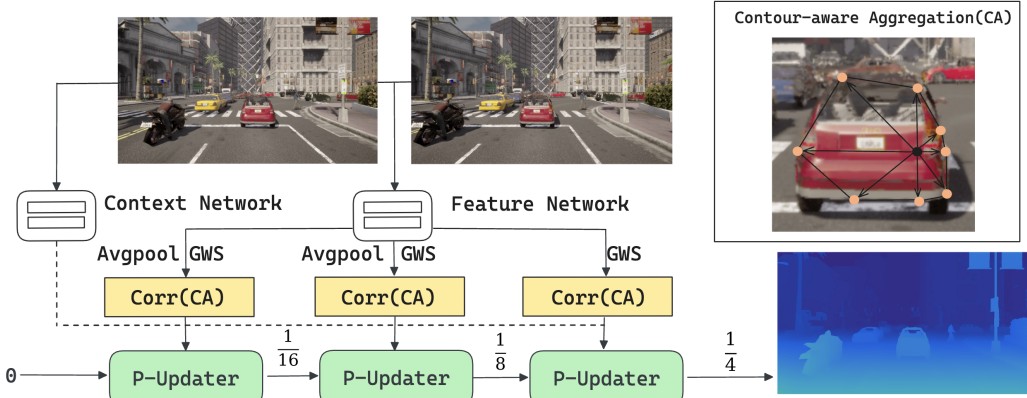

Figure 3: An overview of our proposed method. For a pair of stereo images, they will be fed into the feature network and the context network to generate the multi-level correlations and context feature. The correlations will go through the grouped window shifting (GWS) and contour-aware aggregation (CA) to improve accuracy. Then the method will frequently updates the disparity from the beginning of the zero initialization with propagation updater (P-Updater) which use two different candidate searching methods to improve disparity.

gation, disparity calculation, and disparity optimization. The classic algorithm SGM (Hirschmüller, 2005) selects the disparity of each pixel to form a disparity map, sets a global energy function related to the disparity map, and minimizes this energy function to achieve the goal of finding the optimal dispersion for each pixel. Additionally, some other classic algorithms do not strictly follow the above steps. In PatchMatch Stereo (Bleyer et al., 2011), it continuously iterates to optimize the initial disparity map. In each iteration, each pixel exchanges its disparity value with its neighboring pixels for new cost calculations and retains the disparity value with the lowest cost as its disparity value. However, these algorithms generally have poor parallelism and cannot meet the accuracy and time requirements in practical scenarios when processing high-resolution images.

## 2.2 LEARNING-BASED METHODS

When deep neural networks were first used in the field of stereo matching, they demonstrated significant advantages. In the beginning, this learning-based method was mainly used for feature extraction and matching cost calculation of images. DispNet (Mayer et al., 2016a) concats image pair into a series of convolution operations, while its correlation version DispNetC first performs feature extraction on each image, calculates the correlation, and then performs multi-layer convolution operations. Subsequently, deep neural networks were incorporated into other traditional algorithm steps. IResnet (Duta et al., 2021) introduced a residual layer structure during the disparity optimization phase. AAnet (Xu & Zhang, 2020) introduces deformable convolution in calculating matching cost and cost aggregation. Stereonet (Khamis et al., 2018) uses multi-stage hierarchical refinement from coarse to fine, making the network more lightweight while maintaining good accuracy. Hitnet (Tankovich et al., 2021) has added a slanted window mechanism called tile, which allows the disparity to have two gradients and achieves sub-pixel level accuracy. DeepPruner (Duggal et al., 2019) originates from PatchMatch Stereo (Bleyer et al., 2011) which randomly initializes and propagates within neighbors for narrow cost volume correction. Recently, iterative optimization-based methods have dominated the entire field. Inherited from the optical flow network RAFT (Zhang et al., 2024), RAFT-Stereo (Lipson et al., 2021) constructs a massive cost volume for all relationships between two images called all pair correlation (APC$\in \mathbb{R}^{\mathbf{B}\times\mathbf{H}\times\mathbf{W}\times\mathbf{W}}$) which represents $\mathbf{W}$ matching relationships for $\mathbf{B} \times \mathbf{H} \times \mathbf{W}$ points in the reference image . Moreover, it uses ConvRNN units which gradually optimize from the zero initial states. Subsequently, many improved methods based on this approach emerged and continued to optimize the accuracy. DLNR (Zhao et al., 2023) holds detailed information in feature maps using a decoupled Long Short-Term Memory (LSTM) and achieves remarkable performance. CREStereo (Li et al., 2022) designs a coarse-to-fine network and a special stacked cascaded architecture for inference to improve accuracy. IGEV-Stereo (Xu

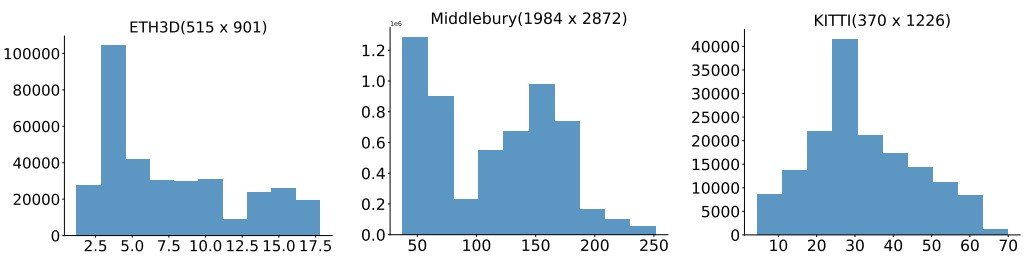

Figure 4: Distribution map of disparity truth maps for three real-world datasets. Obviously, the candidate searching area is much smaller than the width of the image, only $10\% - 30\%$ of it.

et al., 2023a) constructed an additional cost volume and used the WTA method to obtain the initial disparity. Additionally, the cost volume will combine with APC to obtain more accurate cost values.

### 2.3 GENERALIZATION OF STEREO MATCHING

In the absence of massive binocular real datasets, how to make models trained on a large number of simulation datasets perform well in real scenes is an important issue. Depth anything (Yang et al., 2024) uses a data engine approach to train a teacher model with real datasets and then let the model produce predictions for a large number of samples without truth maps to train a student model. Finally, extraordinary accuracy and generalization are achieved. Adastereo (Song et al., 2021) attempts to normalize the cost volume and designs a targeted loss function to solve the problem of domain adaptation. MADNet (Lan et al., 2021) only updates specific modules during adaptive learning, keeping the network constantly in a training state, further improving model accuracy and speed.

## 3 METHOD

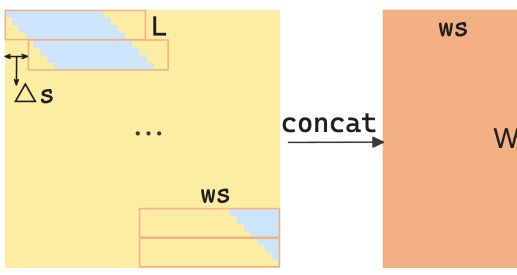

Figure 5: The illustration of grouped window shifting (GWS). The left is the volume containing all the matching relationships of a row of pixels, the correlations of $\mathbf{W} \times \mathbf{W}$ matching relationships. Note that, the blue area is the valid candidate area. The right is the correlation volume that removes most invalid candidate points by setting a window size ($ws$) much smaller than $\mathbf{W}$.

In this section, we present the structure of GCAP-Stereo. It consists of several parts: a feature extractor, a group window shifting mechanism (GWS), a contour-aware correlation aggregation volume (CA), a multi-level iterative updater with Propagation (P-Updater) and an upsampling module.

### 3.1 FEATURE EXTRACTOR

The feature extraction network is consistent with RAFT-Stereo (Lipson et al., 2021). There are two components in the extractor: a context network that extracts multi-scale contextual features for updating the hidden states of ConvGRUs and a feature network that extracts multi-scale features used for constructing the correlation volume. For **context network**, it consists of several residual blocks and downsampling layers and outputs multi-scale context features with designed channels. Then we can get target hidden and context features of the reference image with $\mathbf{tanh}(\cdot)$ and $\mathbf{relu}(\cdot)$. For **feature network**, Given the left and the right images $I_{l/r} \in \mathbb{R}^{3 \times \mathbf{H} \times \mathbf{W}}$, several additional residual blocks will be utilized to generate the feature map at 1/4 of the original size. Note that, the multi-scale (1/8,1/16) of features are implemented by some $\mathbf{avgpool}(\cdot)$ operations which can be represented by $F_{l/r,i} \in \mathbb{R}^{\mathbf{C}_i \times \frac{\mathbf{H}}{i} \times \frac{\mathbf{W}}{i}}$ ($i = 4, 8, 16$ and $\mathbf{C}_i$ for designed channels).

## 3.2 GROUP WINDOW SHIFTING

Although the APC designed by RAFT-Stereo (Lipson et al., 2021) contains amounts of information, many matching relationships will never be used in iterative updates. As shown in fig. 4, for an APC$\in \mathbb{R}^{B \times H \times W \times W}$, only about $10\% - 30\%$ of the matching relationships are potential candidate matching points while other matching relationships may potentially affect the accuracy of the method and occupy unnecessary memory. The grouped window shifting mechanism (GWS) is illustrated in the fig. 5. Specifically, for an APC, we set a designed window size $ws$ and group size $L$ that can contain all potential matching relationships and discard the vast majority of useless information. The shifting mechanism can be formulated as:

$$\Delta s = (\mathbf{W} - ws)L/((\mathbf{W} - \mathbf{D})) \tag{1}$$

where $\mathbf{W}$ is the width of image and $\mathbf{D}$ is the preset max disparity. For each group which contains $L$ rows, we select $ws$ elements to be candidate points. After selecting each group, the selected window will be shifted by $\Delta s$ pixels until it reaches the margin. Note that $ws$ will be set to a value larger than $d$ (usually 1.5 or 2 times) to ensure that potential candidates will not lost.

## 3.3 CONTOUR-AWARE CORRELATION AGGREGATION VOLUME

In order to obtain a more accurate cost volume, we are inspired by SGM (Hirschmüller, 2005) cost aggregation and perform contour-aware cost aggregation on the correlation volume. Specifically, for the cost aggregation formula of SGM (Hirschmüller, 2005):

$$\begin{aligned} L_r(\mathbf{p}, d) = C(p, d) + \min(&L_r(p-1, d), \\ &L_r(\mathbf{p}-1, d-1) + \mathbf{P}_1, \\ &L_r(\mathbf{p}-1, d+1) + \mathbf{P}_1, \\ &\min_i L_r(\mathbf{p}-1, i) + \mathbf{P}_2) \ [(L_r(1, d) = C(1, d))] \end{aligned} \tag{2}$$

which is a dynamic programming equation that simulates two-dimensional cost aggregation by calculating the same distance in multiple one-dimensional directions. $L_r(p, d)$ is the cost along a path traversed in the direction $r$ of the pixel p at disparity d and $\mathbf{P}_1$ and $\mathbf{P}_2$ are the penalities of choosing other disparity. The aggregation cost is the sum of paths which is:

$$S(d) = \sum_{r \in R} L_r(\mathbf{p}, \mathbf{d}) \tag{3}$$

Considering parallelism and accuracy, we performed approximate calculations on it. We assume that when calculating in one direction, if the path is in the same object contour, the same disparity will always be chosen and these disparities often have approximate cost. Therefore, we approximate the value of the points along the path with the cost value of the starting point. Moreover, inspired by CenterNet (Zhou et al., 2019), a method applied to object detection, we aim to make the path in each direction learnable to touch the contour of object illustrated in fig. 3. At this point, we only need to calculate the value of the path endpoint once, which can be formulated as:

$$L_r(d) = aC(1, d_s) + \min_i C(\mathbf{p}, i_e) + \mathbf{P}_2 \tag{4}$$

where $d_s$ is the disparity of the original point and $i_e$ is the disparity of the end point. However, it needs to perform such a calculation on all the disparities of each pixel which is still time-consuming. We observed that the approximated dynamic equation is only related to the values of the starting point and the endpoints in various directions. We consider introducing deformable convolution to directly perform the calculation which can be represented by:

$$L(d) = \sum_{i \in D} (w_1 C(1, i_s) + w_2 C(\mathbf{p}-a, i_e)) + \mathbf{P}_2 \tag{5}$$

It improves parallelism and makes the selection of path aggregation more learnable rather than a simple min operation, resulting in a more accurate and robust aggregation cost.

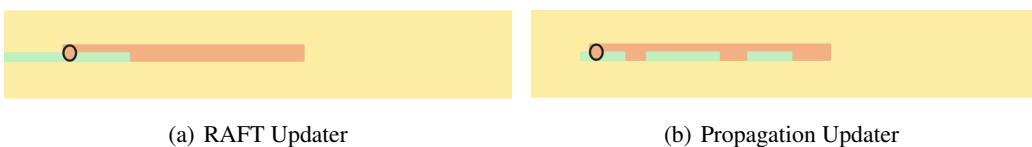

(a) RAFT Updater        (b) Propagation Updater

Figure 6: Two different candidate point searching methods. The RAFT updater always selects several points around for searching, without considering the provided image information which is not efficient. On the contrary, the propagation updater focuses more on local features and selects candidate points in a more targeted way by propagating with its neighbors.

### 3.4 MULTI-LEVEL PROPAGATION UPDATER

For previous RAFT-Stereo based work, the procedure on multiple iterations to optimize disparity can be summarized as the following formula,

$$\Delta f, h_{i+1} = R(f; cor; h_i) \tag{6}$$

$$f' = f + \Delta f \tag{7}$$

where $f$ is the disparity of the current state, $corr$ is the correlation matrix obtained based on its searching in the correlation pyramids, and $h_i$ is the current hidden layer. Based on these parameters, a new hidden layer $h_{i+1}$ and the corresponding increment of flow will be output through the RNN network $R(\cdot; \cdot; \cdot)$. By continuously changing the flow and generating new hidden layers, the disparity will be iteratively optimized. As discussed above, we consider that such a single iteration is not efficient, mainly due to the selection of searching points. Therefore, inspired by PatchMatch Stereo (Bleyer et al., 2011), we have introduced a new iterative update method called propagation updater (P-Updater). As illustrated in fig. 6, if the orange area is the valid candidate point area, for each point, RAFT updater (Zhang et al., 2024) will select several points on both sides, namely the green area, for searching. This not only lacks perception of image information, but also makes it easier to obtain many invalid candidate points. Contrarily, the propagation updater directly propagates with neighbors, searching within a small range of these neighboring points, which can be more targeted and reduce the possibility of selecting invalid candidate points. Considering that multiple iterations of a single level can easily trap the updater in local optima, we adopted a multi-level update approach, where each scale is swapped with different neighbors and two types of updaters are utilized alternately for updates. Finally, the first two low levels updaters use standard bilinear interpolation for upsampling, while the final 1/4 resolution employs the convex combination in RAFT-Stereo (Lipson et al., 2021).

### 3.5 INFERENCE OPTIMIZATION FOR VIDEO STREAM

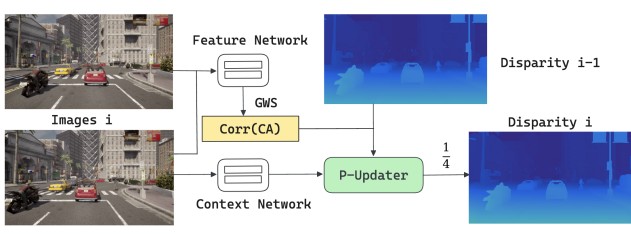

Figure 7: Inference optimization (IO) for the video stream. For the $\mathbf{i_{th}}$ pair of images, we can directly use the previous result (such as the $\mathbf{i - 1_{th}}$ disparity map) as the initial value, and then perform the 1/4 level update directly, which can greatly improve the frame rate without losing accuracy.

As is discussed in previous sections, during training we use a three-level updater at different resolutions to do coarse-to-fine refinement. However, in the inference of video streams in practical scenarios, our method does not need to start from rough results every time. Inspired by Patchmatch Stereo (Bleyer et al., 2011), the results between frames have a strong correlation. Therefore, we choose the results of the previous frame of the video stream as the initial result of the current image, skip coarse-grained optimization and directly perform fine-grained operation. The procedure is interpreted in fig. 7. After starting for a period of time, our method will not start from zero initialization with 1/16- and 1/8-level updates, but instead directly use the previous results to optimize directly with 1/4-level updates. our method will skip the 1/16- and 1/8- level

updaters, and instead use the previous results to make subtle adjustments using the 1/4 level updater each time. This can greatly improve our speed without sacrificing accuracy.

### 3.6 LOSS FUNCTION

Due to our multi-level results in training, the loss function has been modified accordingly. After obtaining intermediate results of 1/16, 1/8, and 1/4 of results , we will upsample the image size to full resolution by bilinear interpolation for the first two levels of the updater, while for the last level of the updater, additional learning will be done through learnable upsampling to full resolution. The exponentially weighted L1 distance will be used in the training with $\gamma$ set to 0.9. Given ground truth $d_{gt}$ and the $l_{th}$ level upsampling prediction $d_{li}$, the total loss is defined as:

$$\mathcal{L} = \sum_{l \in \frac{1}{16}, \frac{1}{8}, \frac{1}{4}} \sum_{i=1}^{n} \gamma^{n-i} ||\mathbf{d}_{gt} - d_{li}||_1 \tag{8}$$

## 4 EXPERIMENTS

### 4.1 DATASETS AND EVALUATION METRICS

Following previous works, we evaluate our method on three common public benchmarks. **Middlebury** dataset is a high-resolution stereo dataset consisting of 23 image pairs captured under various lighting conditions with large-baseline stereo cameras, with disparities reaching up to 600 pixels. It includes 15 training pairs and 15 testing pairs of indoor scenes, providing a challenging benchmark for stereo matching algorithms. **KITTI 2012** and **KITTI 2015** datasets are real-world driving scene datasets consisting of wide-angle stereo image pairs of street views, with sparse disparity ground truth from lidar data. KITTI 2012 contains 194 training and 195 testing pairs, while KITTI 2015 provides 200 training and 200 testing pairs. **ETH3D** dataset consists of gray-scale stereo image pairs with laser-scanned disparity ground truth, featuring a mix of 27 training pairs and 20 testing pairs for both indoor and outdoor scenes, which provides a valuable benchmark for stereo vision tasks in diverse environments.

### 4.2 ZERO-SHOT GENERALIZATION

| method | KITTI-15 | Middlebury Q | Middlebury H | ETH3D |
|---|---|---|---|---|
| SGM (Hirschmüller, 2005) | 23.8 | 10.7 | 25.2 | 12.9 |
| PatchMatch Stereo (Bleyer et al., 2011) | 27.3 | 16.1 | 38.6 | 24.1 |
| HD3 (Yin et al., 2019) | 26.5 | 18.1 | 34.2 | 30.1 |
| PSMNet (Chang & Chen, 2018) | 16.3 | 14.2 | 25.1 | 23.8 |
| DSMNet (Zhang et al., 2020) | 6.5 | 8.1 | 13.8 | 6.2 |
| GANet (Wang, 2022) | 11.7 | 11.2 | 20.3 | 14.1 |
| RAFT-Stereo (Lipson et al., 2021) | 5.74 | 9.36 | 12.59 | 3.28 |
| IGEV-Stereo (Xu et al., 2023a) | 6.8 | 6.2 | 7.1 | 3.6 |
| Selective-IGEV (Wang et al., 2024) | 6.31 | 5.33 | **7.03** | 4.17 |
| GCAP-Stereo(Ours) | **5.68** | **4.93** | 7.21 | **1.52** |

Table 1: Zero-shot generalization experiments. All methods were sorely trained on Scene-Flow (Mayer et al., 2016b) and directly tested on the KITTI2015 (Geiger et al., 2012), Middlebury (Scharstein et al., 2014) quarter and half, and ETH3D (Schöps et al., 2017) validation datasets. The values are the percent of pixels that EPE scores larger than a specified value. In this table, it is set as bad 3.0 for KITTI, bad 2.0 for the Middlebury quarter and half and bad 1.0 for ETH3D.

Firstly, we focused on evaluating the zero-shot generalization ability of GCAP-Stereo from synthetic training data to unseen real-world datasets. Due to the current difficulty of binocular camera cali-

| Model | P-Updater | Multi-level | CA | GWS | Bad 1.0(32) | Bad 1.0(8) | epe |
|---|---|---|---|---|---|---|---|
| Baseline | - | - | - | - | 8.32 | 12.66 | 0.27 |
| P | ✓ | - | - | - | 6.55 | 7.35 | 0.22 |
| P+M | ✓ | ✓ | - | - | 6.44 | 7.11 | 0.18 |
| P+M+C | ✓ | ✓ | ✓ | - | 4.52 | 5.28 | 0.15 |
| GCAP-Stereo | ✓ | ✓ | ✓ | ✓ | **4.47** | **4.91** | **0.14** |

Table 2: Ablation study of proposed method on the sceneflow validate set. The baseline is RAFT-Stereo and the table shows the two bad 1.0 metrics after 8 iterations and 32 iterations.

bration, there is no large-scale real dataset available, making this ability crucial to the field of stereo matching. In this experiment, we trained on a simulation dataset sceneflow (Mayer et al., 2016b) for 200000 steps and directly verified its performance on three real datasets, as shown in Table 1. Our method has demonstrated absolute advantages on various datasets, especially on ETH3D (Schöps et al., 2017), with the zero-shot performance alone surpassing the performance of networks such as RAFT-Stereo, GMStereo, and HITNet fine-tuning on eth3d.

## 4.3 ABLATION STUDY

In this subsection, we mainly focus on the influence of the proposed methods on the accuracy and inference time. All training procedures are held on the sceneflow dataset with 200000 steps and a learning rate of 0.0002. Eventually, different models will be validated on the ETH3D validation set. Note that, the baseline used in this experiment is RAFT-Stereo (Lipson et al., 2021).

**Effectiveness of Multi-level P-Updater.** To compare the performance of P-updater, we first performed a single-level updater replacement. As shown in table 2, by simply improving the search method of RAFT-Stereo, the accuracy was significantly improved. This is because the searching method of P-Updater is more related to the image information, making the final updated results undoubtedly more ideal. Furthermore, we transformed the single -evel P-Updater into a multi-level updater while maintaining the same number of iterations. As shown in table 1, even with the use of a lighter coarse-grained updater, the accuracy did not decrease but instead increased. This is because the propagation method at multi-level are different, and compared to a single-level fixed propagation method, the receptive field is larger, which can make it easier for the final result to be out of local optima and achieve better results.

**Effectiveness of CA and GWS.** As shown in the table 2, by approximating the SGM equation and using deformable convolution for calculation, the accuracy can be further improved with little cost. This is because the CA is not limited to small-area neighbor convolutions, but rather extends as much as possible around the object contour boundary like CenterNet (Zhou et al., 2019), resulting in more abundant information obtained through aggregation. Furthermore, GWS can greatly reduce the volume of the correlation, thereby reducing the computational cost brought by CA and make the searching area more concentrated and targeted. The combination of the two makes our method more effective.

**Effectiveness of single optimization.** Our method can still perform well at a low number of iterations. As shown in table 2, we report the bad 1.0 score with different iterations. In the case of only 8 iterations, the accuracy of RAFT-Stereo will sharply decrease, but our method can still maintain stability, indicating that our single iteration is more efficient and robust. Moreover,in just 8 iterations, even with the addition of only one single-level P-Updater to our method, it has already surpassed RAFT-Stereo, which has undergone 32 iterations.

## 4.4 COMPARISONS WITH STATE-OF-THE-ART

**Middlebury.** Unlike previous works which leverage multiple datasets to finetune, we only finetune our Scene Flow pre-trained model on the mixed InStereo2k and Middlebury datasets using a crop size of $384 \times 768$ with a batch size of 8 for 100k steps. We then adopt 2-stage inference to evaluate our method on the test set at $1536 \times 2048$ using resized full-resolution images. As shown in table 3,

|  | ETH3D | | | Middlebury | | |
|---|---|---|---|---|---|---|
|  | Bad 1.0 | Bad 2.0 | EPE | Bad 1.0 | Bad 2.0 | Bad 4.0 |
| HITNet (Tankovich et al., 2021) | 2.79 | 0.80 | 0.20 | 13.30 | 6.46 | 3.81 |
| RAFT-Stereo (Lipson et al., 2021) | 2.44 | 0.44 | 0.18 | **9.37** | 8.07 | 2.75 |
| CroCo-Stereo (Weinzaepfel et al., 2022) | 0.99 | 0.39 | **0.14** | 16.90 | 7.29 | 4.18 |
| AdaStereo (Song et al., 2021) | 3.09 | 0.65 | 0.25 | 29.50 | 13.70 | 6.35 |
| GMStereo (Xu et al., 2023b) | 1.83 | 0.25 | 0.19 | 23.60 | 7.14 | 2.96 |
| IGEV-Stereo (Xu et al., 2023a) | 1.12 | **0.21** | **0.14** | 9.41 | 4.83 | 3.33 |
| GCAP-Stereo(ours) | **0.95** | 0.24 | **0.14** | 10.00 | **4.31** | **2.46** |

Table 3: Quantitative results on ETH3D and Middlebury benchmark.

our GCAP-Stereo surpasses the published state-of-the-art by 10.77% on the bad 2.0 metric, 10.55% on the bad 4.0 metric, and ranks 3[rd] place on the bad 1.0 metric.

**ETH3D.** Unlike Selective-Stereo which applies a mixed dataset of CREStereo, InStereo2k and ETH3D to finetune for 90k steps, we only finetune our Scene Flow pre-trained model on the mixed InStereo2k and ETH3D datasets using a crop size of $384 \times 512$ with a batch size of 8 for 20k steps. We evaluate our method on the test set with the size of $768 \times 1024$ where 2-stage inference is adopted. We achieve the 1[st] place on the majority of the metrics among all published methods, surpassing the published state-of-the-art by 4.04% on the bad 1.0 metric. Our GCAP-Stereo ranks 2[nd] place on the bad 1.0 metric and 1[st] place on EPE metric respectively. Quantitative comparisons are tabulated in table 3.

**KITTI.** We fine-tune the model for another 50K iterations on KITTI 2012 and 2015 training sets. The initial learning rate is set to 0.0001. Finally, we achieve competitive performance on both datasets and show a visual comparison of KITTI 2015 in fig. 8.

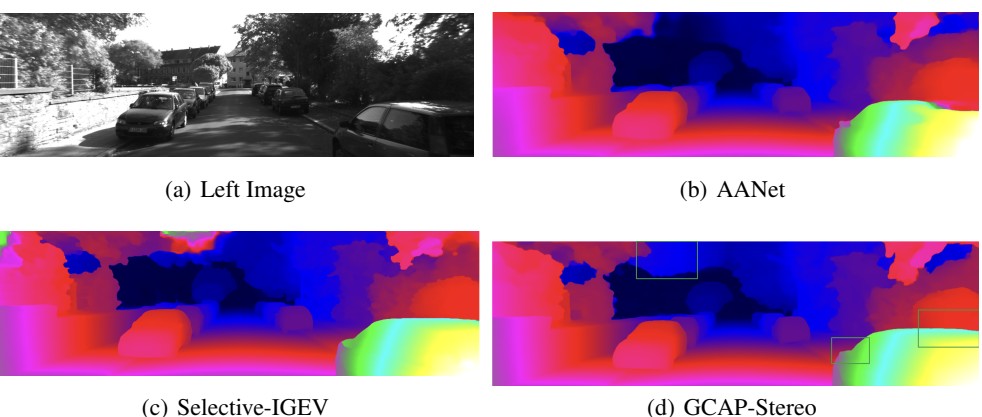

|  |  |
|---|---|
| (a) Left Image | (b) AANet |
| (c) Selective-IGEV | (d) GCAP-Stereo |

Figure 8: Visual comparisons with other methods on the case of KITTI 2015 leaderboard. Our method performs better with less distortion and incorrect matching illustrated by green boxes.

### 4.5 INFERENCE FOR VIDEO STEAM

To verify the feasibility of our inference optimization on video streams, we chose to conduct our simulation experiments on the CARLA simulator (Dosovitskiy et al., 2017), an autonomous driving simulator. This simulator can not only output real-time disparity maps, but it also can perform automatic navigation at any selected location. Specifically, we will conduct experiments directly on the simulator using the baseline and GCAP-Stereo trained on the sceneflow dataset. Each method will run on the same road for the same time, and finally calculate the average result of all obtained disparity maps during this period. The result is shown in table 4. It proves again that our method has

better ability of the zero-shot genealization and the inference optimization of video streams does not lead to a decrease in accuracy but can bring huge speed improvement.

|  | Bad 2.0 | EPE | Time(frames/s) |
|---|---|---|---|
| RAFT-Stereo | 13.07 | 2.67 | 6 |
| GCAP-Stereo | **9.35** | **1.73** | 8 |
| GCAP-Stereo with IO | 9.41 | 1.75 | **15** |

Table 4: Performance and frame rate comparison in CARLA simulator

### 4.6 Practical performance

To further validate the generalization ability of our model, we will transfer the model trained on the sceneflow dataset to real-world scenarios for testing. Specifically, we have built a simple demo and will conduct a visual verification using the Jetson Orin Nano (Süzen et al., 2020) and a designed binocular camera system. The visualization results are shown in fig. 9 and illustrated that our method has a more global accuracy optimization for the overall contour of the object and the small gaps between objects, resulting in a significant improvement in the disparity accuracy of the object, as shown in the red "0" and the several gears in the set of images.

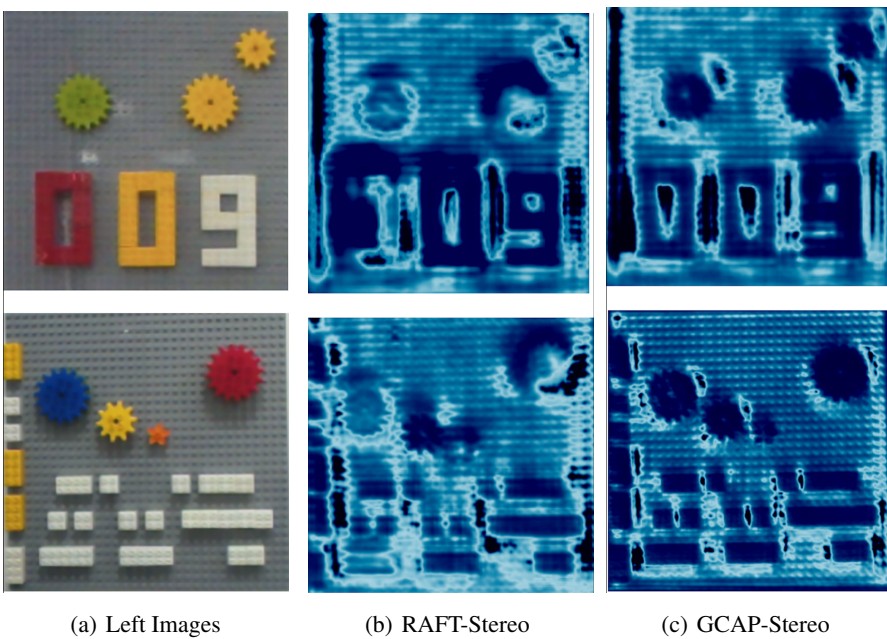

(a) Left Images      (b) RAFT-Stereo      (c) GCAP-Stereo

Figure 9: Visialization comparison of real-world demo between RAFT-Stereo and GCAP-Stereo.

## 5 Conclusion

To solve the common problems in iterative networks for stereo matching, we propose Grouped Correlation Aggregation with PropagationI (GCAP-Stereo), a new solution for stereo matching. The efficiency of single iteration optimization has been improved by introducing a new propagation-based updater. Through improving traditional algorithms, targeted modifications have been made to the correlation volume to make it more robust and accurate. Finally, targeted optimization was carried out on the inference of the video stream. GCAP-Stereo ranks 1st on ETH3D two-view stereo benchmarks and achieves competitive performance on KITTI 2012/2015 and Middlebury among published methods. Moreover, our method demonstrates excellent performance advantages in video stream testing and zero-shot generalization, which has superior cross-domain generalization and real-time performance. We believe our work will be an important technique empowering the high-precision binocular vision system.

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
