# OpenReview forum: "Grouped Correlation Aggregation with Propagation for Stereo Matching"
_ICLR.cc/2025/Conference — ICLR 2025 Conference Withdrawn Submission_

### Official Review · Reviewer_8DB1 · 2024-10-29

**Soundness:** 2
**Presentation:** 2
**Contribution:** 2
**Rating:** 5
**Confidence:** 4

**Summary:**

This work introduces a new stereo matching method designed to address inefficiencies in existing iterative optimization-based stereo approaches. On the top of RAFT-Stereo, a new iterative updater based on PatchMatch Stereo is presented for better convergence. Also, the contour-aware aggregation (CA) was proposed by modifying multi-pass dynamic programming used in the semi-global matching (SGM). The grouped window-shifting (GWS) is further presented by considering that a large portion of matching costs are never accessed in the stereo matching process. For a faster inference on video stream, authors introduce a simple trick of using previous frame's estimate as an initial disparity at current frame.

**Strengths:**

1) The deformable convolution for approximating dynamic programming-based cost aggregation and the multi-level propagation updater seem to be effective according to experimental results.

**Weaknesses:**

1) The contour-aware aggregation (CA) needs clarifications.
The original dynamic programming formulation of (2) is approximated as (4), when starting point and endpoint of the same object contour are given. Rather than predicting them exhaustively, authors propose to use deformable convolution. Though the overall idea is understandable, this part should be revised in a completely different form. In (4), what does '1' means? This may be a starting point $p_s$. In (5), 'a' seems to be a pixel of deformable convolution, but it is not defined correctly. Are $w_1$ and $w_2$ learnable?

2) Multi-level propagation updater in Section 3.4 needs more details.
- $h_i$ should be explained for completeness.
- Figure 6: what do $x$ and $y$ dimensions mean? And why is the green area isolated with multiple parts?
- PatchMatch Stereo uses current estimated disparities of neighboring pixels as disparity candidates during an iterative optimization. Is a similar method used in the proposed method?
- What does the multi-level update mean? Are disparity candidates used from different resolutions?
- 'each scale is swapped with different neighbors': This is hard to understand.
- 'two types of updaters are utilized alternately for updates.': The two types of updaters have not been explained before.


In summary, this work introduces an efficient iterative optimization method, which is based on RAFT-Stereo. The deformable convolution for approximating dynamic programming-based cost aggregation and the multi-level propagation updater seem to be effective according to experimental results. Nevertheless, substantial revisions are required in terms of technical presentation.

**Questions:**

See the comments in the weaknesses.

**Details Of Ethics Concerns:**

N.A.

---

### Official Review · Reviewer_MUs3 · 2024-11-01

**Soundness:** 2
**Presentation:** 2
**Contribution:** 2
**Rating:** 3
**Confidence:** 4

**Summary:**

This is a paper based on RAFT Stereo to improve binocular reconstruction performance, training speed, and memory consumption. The main improvements made are: 1. Replace the original RNN with PatchMatch as the new updator to improve the performance of a single iteration. 2. The window shifting mechanism was used to reduce the volume of the cost volume. 3. SGM based cost aggregation is used to enhance the robustness of the cost volume.

**Strengths:**

1. Conducted experiments on diverse datasets.
2. The method has improved the performance of the baseline.

**Weaknesses:**

1. The window shifting mechanism is based on the continuity assumption of scene disparity or depth, and then sets a threshold to crop the cost volume. This is more like a trick, rather than an innovative module. And this article does not explain the potential negative impacts that this may bring.
2. Lack of novelty in SGM based cost aggregation. The method of cost aggregation has been widely and persistently applied in geometric estimation tasks. However, since this paper has not made many improvements to it, we believe that this cannot be considered as the contribution point of this paper.
3. This article lacks an explanation for changing the original RNN uploader to PatchMatch.
4. The explanation of some legends is not clear enough. For example, in Fig 6, it is not explained what the different colored bands represent. Although it is mentioned in the main text, it caused difficulties in reading.
5. The performance on the actual benchmark differs from the performance described in the paper. Line 454 "Our GCAP Stereo ranks 2 nd place on the bad 1.0 metric and 1 st place on EPE metric respect", but when we checked the actual benchmark, we found that the method is not the first. And there is a lack of comparison with the latest methods, such as CVPR2022 CREStereo and others, CVPR2024 LoS: Local Structure guided Stereo Matching, Confidence Aware Stereo Matching for Realistic Cluttered Scenarios. ICIP 2024 also includes a large number of methods that have not been publicly published as papers. Overall, we believe that ranking first is an exaggeration.

**Questions:**

1. Why replace the RNN updator with PatchMatch updator, and what are the core advantages?
2. Will cutting the cost volume result in performance loss?

---

### Official Review · Reviewer_496Q · 2024-11-03

**Soundness:** 2
**Presentation:** 1
**Contribution:** 2
**Rating:** 3
**Confidence:** 4

**Summary:**

The paper presents grouped correlation aggregation with propagation to improve the performance of single iteration optimization. Specifically, the author has proposed some novel designs based on traditional methods, such as a novel updater based on PatchMatch Stereo, a modified SGM-based cost aggregation, and grouped window-shifting mechanism.

The author claims that the proposed method outperforms existing methods on public benchmarks such as ETH3D and demonstrates advantages in zero-shot generalization and video stream inference.

**Strengths:**

The motivation is commendable; this paper aims to improve the optimization efficiency of RAFT-Stereo. The author believes that the all-pairs correlation constructed by RAFT-Stereo contains a large amount of redundant information, and therefore introduces the grouped window-shifting mechanism to significantly reduce the cost volume.

Furthermore, considering the matching cost between single pixels, which often leads to frequent occurrences of noise points and matching errors, the author proposes a modified SGM-based cost aggregation method to improve the robustness of the cost.

The author provides a distribution map of disparity ground truth (gt) to give an intuitive explanation.

**Weaknesses:**

1. The proposed solution seems to be not very novel and effective. In fact, IGEV-Stereo also constructs a geometry encoding volume for a small range of disparities (disp < 192 px), filtering out most of the redundant disparities.

2. This performance comparison is quite limited, and the author has missed comparisons with recent methods on both the KITTI benchmark and the Middlebury benchmark, such as Selective-IGEV, CREStereo, Los.

3. The writing quality is poor, and the method description is unclear. The author has not adequately explained how the propagation updater works in Section 3.4.

4. The results on Scene Flow are not convincing. In fact, previous methods typically use the test set to evaluate model performance, while the author only conducts ablation studies on the validation set, which is not convincing.

**Questions:**

1. Compared with the geometry encoding volume of IGEV-Stereo, what are the advantages of the grouped window-shifting mechanism in this paper?

2. Please provide the evaluation results on the KITTI benchmark

3. Please include more comparisons with recent methods, such as Selective-IGEV [CVPR 2024], CREStereo [CVPR 2022], Los[CVPR 2024] on the Middlebury benchmark.

4. Please provide a comprehensive comparison with RAFT-Stereo when the number of iterations increases from 1 to 16, so that readers can have a more intuitive understanding.

---

### Official Review · Reviewer_wSKi · 2024-11-03

**Soundness:** 3
**Presentation:** 3
**Contribution:** 3
**Rating:** 6
**Confidence:** 4

**Summary:**

This paper proposed a new propagation-based updater, which combines the grouped window shifting mechanism and a contour-aware aggregation modified from semi-global matching (SGM), the proposed GCAP-STEREO has superior cross-domain generalization and real-time performance.

**Strengths:**

1. This paper introduces the grouped window-shifting mechanism to discard the vast majority of invalid points for efficiency and proposes a modified SGM-based cost aggregation method for robustness. GCAP-Stereo ranks 1st on ETH3D and has superior cross-domain generalization and real-time performance.

**Weaknesses:**

1. The performance comparison is insufficient. Why is there no comparison of performance on the KITTI leaderboard? KITTI is currently the most common public benchmark available. The Zero-shot generalization comparison in Table 1 also lacks the comparison in KITTI 2012.
2. The comparison method of generalization is not sufficient. In my opinion, to prove the generalization of this paper, you should compare some algorithms known for generalization, such as Cfnet: Cascade and fused cost volume for robust stereo matching and CREStereo: Practical stereo matching via cascaded recurrent network with adaptive correlation, only such a sufficient contrast can be convincing enough.

**Questions:**

1. Can you provide a performance comparison on the KITTI leaderboard and a generalization comparison on the KITTI 2012? Adequate experimental comparisons are crucial for enhancing the persuasiveness of the results.
2. Can you provide some comparison to the SOTA solution for generalization? The comparison with the precision scheme may not be convincing enough.

---

### Note · Authors · 2024-11-15

I have read and agree with the venue's withdrawal policy on behalf of myself and my co-authors.